

# Effect of grassland cutting frequency on soil carbon storage - A case study on public lawns in three Swedish cities

**C. Poeplau[1], H. Marstorp[2], K. Thored[1] and T. Kätterer[1]**

[1]Swedish University of Agricultural Sciences (SLU), Department of Ecology, Box 7044, 75007 Uppsala, Sweden

[2]Swedish University of Agricultural Sciences (SLU), Department of Soil and Environment, Box 7014, 75007 Uppsala, Sweden

Correspondance: C.Poeplau (Christopher.poeplau@slu.se)

## Abstract

Soils contain the largest terrestrial carbon pool and thus play a crucial role in the global carbon cycle. Grassland soils have particularly high soil organic carbon (SOC) stocks. In Europe (EU 25), grasslands cover 22% of the land area. It is therefore important to understand the effects of grassland management and management intensity on SOC storage. City lawns constitute a unique study system in this context, since they provide a high diversity and wide range of functionalities and thus management intensity per unit area. In this study we investigated frequently mown (on average 8 times per season) utility lawns and rarely mown (once per season) meadow-like lawns at three multi-family housing locations in each of three Swedish cities, Uppsala, Malmö and Gothenburg. The two different lawn types were compared regarding their aboveground net primary production (NPP) and SOC storage. In addition, root biomass was determined in Uppsala. We found significantly higher aboveground NPP and SOC concentrations and significantly lower soil C:N ratio for the utility lawns compared with the meadow-like lawns. On average, aboveground NPP was 24% or 0.7 Mg C ha$^{-1}$ yr$^{-1}$ higher and SOC was 12% or 7.8 Mg ha$^{-1}$ higher. Differences in SOC were well explained by differences in aboveground NPP ($R^2$=0.39), which indicates that the increase in productivity due to more optimum $CO_2$-assimilating leaf area, leading to higher carbon input to the soil, was the major driver for soil carbon sequestration. Differences in soil C:N ratio indicated a more closed N cycle



in utility lawns, which might have additionally affected SOC dynamics. We did not find any
difference in root biomass between the two management regimes, and concluded that cutting
frequency most likely only exerts an effect on SOC when cuttings are left on the surface.

**1 Introduction**
Soils contain the largest terrestrial carbon pool (Chapin et al., 2009). The balance of soil organic
carbon (SOC) inputs and outputs is therefore critical for the global carbon balance and thus for
the concentration of greenhouse gases in the atmosphere. Globally, 3650 Mha or 68% of the total
agricultural area is under permanent pasture or meadows (Leifeld et al., 2015). In Europe (EU-
25), grassland covers 22% of the land area (Soussana et al., 2007). Grassland soils store among
the highest amounts of SOC, which is primarily related to the high belowground carbon input by
roots and their exudates (Bolinder et al., 2012). Soils rich in SOC are potential hot-spots for $CO_2$
emissions, when a management or land-use change-induced imbalance in carbon input and
output occurs. It is therefore important to understand the effect of management practices on
grassland SOC storage. It has been demonstrated, that the type, frequency and intensity of net
primary production (NPP) appropriation play a crucial role for the carbon balance and SOC
stocks of grassland ecosystems (Soussana et al., 2007). These effects can be diverse, direct or
indirect and can counterbalance each other. One direct management intensity effect on SOC,
which is mediated by grazing, cutting or fertilisation regime, is obviously the change in carbon
input via the degree of biomass extraction and altered photosynthetic activity (Wohlfahrt et al.,
2008a). Furthermore, above- and below-ground allocation patterns may change with cutting
frequency (Seiger and Merchant, 1997). Recently, Leifeld et al. (2015) reported faster root
turnover in moderately and intensively managed alpine grasslands than at less intensively grazed
sites. They concluded that management is a key driver for SOC dynamics and should be included
in future predictions of SOC stocks. Nutrient status, species composition and diversity are highly
management-dependent and interfere with the carbon cycle in several ways, including effects on
the decomposer community and its substrate use efficiency (Ammann et al., 2007; Kowalchuk et
al., 2002). However, to assess management effects on SOC stocks, which are presumably smaller
than land use change effects such as conversion from permanent pasture to arable land (Poeplau
and Don, 2013) and might not be visible in the short term, it is very important to find suitable



study systems with long-lasting strong contrasts in management intensity over a limited spatial scale and with limited soil variability. For agroecosystems, this situation is usually created in long-term experiments, which are designed to study such questions. In a global compilation of all existing agricultural long-term field experiments, only 49 out of >600 experiments are listed as including permanent grassland (pasture or meadow) (Debreczeni and Körschens, 2003). Thus, the current quantitative and mechanistic understanding of grassland management effects on SOC stocks is certainly limited, since existing studies are often strongly confounded by external factors such as elevation gradients (Leifeld et al., 2015; Zeeman et al., 2010). As an alternative to long-term plot experiments, urban areas can be appropriate study systems. Lawns, public green areas and parks are omnipresent in urban areas and are usually managed in a similar way for a long time, so that an approximation to equilibrium SOC stocks can be assumed. Over a comparatively small spatial scale, a great diversity of different functional types of grasslands with different management intensities can be present.

Lawns cover the majority of all green open spaces in urban landscapes (70-75%) according to Ignatieva et al. (2015). It has been estimated that turf grass lawns cover approximately 16 M ha of the total US land area, which in the 1990s was three-fold the area of irrigated maize (Milesi et al., 2005; Qian et al., 2010). Although robust global estimates of the coverage of turfgrass lawns are scarce, these few existing figures indicate the potential importance of lawn management for the global carbon cycle. There is thus a need to quantify the carbon footprint of differently managed lawns, for which SOC is of major importance. The social, ecological and aesthetic values and the total environmental impact of lawns have not been comprehensively evaluated (Ignatieva et al., 2015). However, in the transdisciplinary Swedish LAWN project (http://www.slu.se/lawn), lawns are studied from different perspectives.

In this study we analysed two types of lawn under different management intensity (cutting frequency) associated with multi-family housing areas, which were intensively monitored at three sites in each of three Swedish cities. We examined how cutting frequency affected: i) NPP and SOC, and ii) the mechanisms involved for potential differences in SOC storage.



## 2 Materials and Methods

### 2.1 Study sites

Public lawns in multi-family housing areas were investigated in three different cities, Gothenburg, Malmö and Uppsala, and at three different locations in each city (Table 1). The nine selected multi-family housing areas were established at approximately the same time during the early 1950s. In each area, triplicate plots of two different lawn types were studied: utility lawn and meadow-like lawn, with each plot comprising one complete lawn. The utility lawn was kept short during the year and was mown on average every 18 days within the mowing period, which approximately corresponds with the growing period (May to mid-October in Uppsala, April to late October in Gothenburg and Malmö). The meadow-like lawns were only cut once, or twice in the single case of one lawn in Uppsala (Tuna Backar). Grass cuttings were left on the surface on both lawn types. None of the lawns received any fertiliser. Grass species composition did not differ greatly between the cities, with about 5-10 different grass species abundant in utility lawns and meadow-like lawns. Utility lawns consisted of sparser, low-growing species such as *Poa annua*, *Agrostis capillaris*, *Lolium* spp. and *Festuca rubra*, while the most abundant grass species in meadow-like lawns were *Phleum pratense*, *Alopecurus pratensis* and *Arrhenatherum* spp. (J. Wissman, pers. comm. 2015).

### 2.2 Estimation of aboveground net primary production and root biomass

Aboveground NPP in the utility lawns was estimated by repeated sampling of aboveground biomass after the first mowing in spring by the local authority. Sampling was conducted on average 12±6 days after the lawn was mown. For the meadow-like lawns, biomass was collected on several occasions even before the mowing to determine total growth at that specific time. After the first cut, meadow-like lawns were treated as utility lawns. The plots were sampled at four locations using a 50 cm x 50 cm square frame. Sampling locations were selected to be representative of the total lawn area, so therefore sampling under trees or in proximity to other vegetation was avoided. The harvested biomass was dried at 70°C, weighed and multiplied by 4 to obtain the biomass for 1 m$^2$. The mean of the four replicates was divided by the number of days between the last cutting and sampling to obtain daily growth rate. This growth rate was extrapolated to cover all days between previous sampling and next mowing for which no growth





rate was determined. On average, this period accounted for 7±6 days after each cutting event,
and thus data coverage (time for which the actual growth rate was measured) was more than
82±6% for the period between 1 January and the last sampling date, which was on average on 5
October ±7 days. On the basis of these daily growth rates, we calculated cumulative growth until
the last sampling. Since this day varied slightly between plots and sites, we fitted a simple
vegetation model based solely on the plant response to air temperature, as developed by Yan and
Hunt (1999) to each growth curve in order to determine the regrowth after the last sampling until
the end of the vegetation period. The original equation is:
$$r = R_{max} \left( \frac{T_{max} - T}{T_{max} - T_{opt}} \right) \left( \frac{T}{T_{opt}} \right)^{\frac{T_{opt}}{T_{max} - T_{opt}}},$$    Eq. 1
where $r$ is the daily rate of plant growth, $T$ is the measured temperature at any day, $T_{max}$ is the
maximum temperature (which was set to 30°C in this study), $T_{opt}$ is the optimal temperature
(which was set to 25°C in this study) and $R_{max}$ is the maximal growth rate at $T_{opt}$. Instead of using
$R_{max}$, which is used in eq. 1 to scale the temperature response function to actual observed
maximal plant growth at optimal temperature, we scaled the model by forcing the cumulated r
through the cumulated NPP value on the date of the last sampling, as illustrated in Figure 1 using
the example of the Björkekärr site in Gothenburg. The good fit indicates that: i) the growth
dynamics, and thus absolute growth, were well captured by the method used; and ii) the model
fits provide an unbiased and standardised extrapolation of aboveground NPP for the entire
growing period. Daily mean air temperature values for the closest weather stations of the
Swedish Meteorological Service (SMHI) to Malmö and Gothenburg were downloaded from
http://www.smhi.se/klimatdata. Daily average air temperature values for Uppsala were obtained
from the Ultuna climate station run by the Swedish University of Agricultural Sciences (SLU).
Root biomass was only determined once, and only in Uppsala. In each lawn, four cylindrical soil
cores of 7 cm diameter and 10 cm depth were taken at 0-10 cm soil depth. Aboveground plant
material was removed and soil cores were thoroughly rinsed and then put in a water bucket to
completely separate roots from soil. Roots were dried at 105°C weighed and analysed for carbon
(C) and nitrogen (N) content. Assuming a carbon content of 45% for plant biomass, we were able
to determine and subtract the adhering soil in the weighed root samples mathematically, as
described in Janzen et al. (2002).



### 2.3 Soil sampling, analysis and SOC stock calculation


Soils were sampled in autumn 2014 to a depth of 20 cm using an auger (2.2 cm diameter). In
each plot, 10 randomly distributed soil cores were taken and pooled to one composite sample.
Soils were dried at 40°C, sieved to 2 mm and visible roots were manually removed. Soil pH was
determined in water and samples with a pH value exceeding 6.7 were analysed for carbonates.
Total soil carbon and nitrogen were determined by dry combustion of 1 g of soil using a LECO
TruMac CN analyser (St. Joseph, MI, USA) and carbonate carbon was determined using the
same instrument after pretreatment overnight at 550°C. Organic soil carbon was calculated as the
difference between total carbon and carbonate carbon. Soil bulk density [g cm$^{-3}$] was determined
by taking undisturbed cylindrical soil cores of 7 cm diameter and 10 cm depth to an approximate
depth of 5-15 cm, drying them at 105°C and weighing them. Four samples were taken in each
plot. To account for the fact that SOC stocks under contrasting management regimes should be
compared on the basis of equivalent soil masses (Ellert and Bettany, 1995), we conducted a
simple mass correction in which we first calculated the soil mass (SM) [Mg ha$^{-1}$] of each plot
using the equation:
$$SM = BD \times D \times 100,$$ Eq. 2
where BD is the soil bulk density [g cm$^{-3}$] and D is the sampling depth [cm]. The lower average
soil mass measured at each pair was then used as the reference soil mass (RSM) to which the
other treatment of each pair (three pairs per site) were adjusted.
SOC stocks [Mg ha$^{-1}$] were then calculated using the equation:
$$SOCstock = RSM \times \frac{C}{100}$$ Eq. 3
where C is carbon concentration [%]. At one site in Gothenburg (Kyrkbyn), one pair of lawn
types had a large difference in soil texture, with 15% clay in the utility lawn and 30% in the
meadow-like lawn. The SOC concentration varied by a similar amount (2.46% compared with
4.58%), which was an outlying high difference when compared with that of all other pairs. We
attributed this to differences in soil texture and excluded this pair from the analysis. Apart from
slight differences in soil texture, the basic assumption was that the underlying pedology and
initial soil carbon stocks were similar for both lawn types, or at least not systematically biased.



Differences in soil texture between lawn types at each site was further not correlated to
differences in SOC concentration ($R^2$=0.02).

**2.4 Statistics**

We used linear mixed effect models to analyse the effect of lawn management on aboveground
NPP and SOC concentration and stocks (Pinheiro et al., 2009). Management (utility vs. meadow-
like lawn) was used as the fixed effect, while city and site were used as nested random effects
(site nested in city). Average differences in SOC stocks between the different lawn types at each
site were calculated and related to different explanatory variables, such as average clay content,
differences in clay content between lawn types (absolute and relative), soil pH, mean annual
temperature (MAT), mean annual precipitation (MAP) and differences in aboveground NPP.
Generalised linear models were used for multiple regression analysis. All statistical analyses
were performed with the R software. All values in the text and diagrams represent
mean±standard deviation.

**3 Results**

3.1 Effect of lawn management on net primary production and soil carbon and nitrogen
The intensively managed, i.e. frequently mown, utility lawns produced significantly (p=0.003)
more aboveground biomass (NPP) than the meadow-like lawns, which were cut only once a year
(Figure 2). At seven out of nine sites, NPP was higher in the utility lawns than in the meadow-
like lawns. The difference between the lawn types was most pronounced in Uppsala, where the
average NPP of the utility lawns (4.2±0.9 Mg C ha$^{-1}$) was twice that of the meadow-like lawns
(2.1±0.3). In contrast, two out of three sites in Gothenburg showed higher NPP on the meadow-
like lawns. Across all sites, the NPP of the utility lawns was 24% higher. Total root biomass, as
investigated at the three sites in Uppsala, was not significantly influenced by management
intensity and indicated a smaller ratio of belowground to aboveground NPP in meadow-like
lawns (Figure 3).
Concentrations of SOC were also positively affected by greater cutting frequency. Utility lawns
had significantly higher (p=0.01) SOC concentration than meadow-like lawns (Figure 4). Again,



the difference between the two lawn types was most pronounced in Uppsala, with an average
SOC concentration of 3.9±0.6% in the utility lawns and 2.9±0.9% in the meadow-like lawns. In
both Malmö and Gothenburg, we found one site with higher average SOC concentration in the
meadow-like lawns. The calculated SOC stocks are listed in Table 3. The average SOC stock
difference between the two differently managed lawn types was 7.8 Mg ha$^{-1}$ or 12%. The very
similar patterns observed for the variables NPP and SOC suggest that the SOC changes were
driven by NPP and thus carbon input. In fact, the difference in SOC stock between management
regimes at each site was significantly correlated to the difference in NPP (Figure 5).
Furthermore, difference in SOC stock did also significantly increase with average clay content
($R^2$=0.26), but difference in NPP was even stronger correlated to clay content ($R^2$=0.36).
The soil C:N ratio of the meadow-like lawns (13.2±1.2) was significantly higher (p=0.007) than
that of the utility lawns (12.6±0.7), indicating that the soil organic matter under the utility lawns
was relatively enriched in nitrogen (Figure 6).

## 216   4 Discussion

### 217   4.1 Effect of cutting frequency on aboveground productivity

We showed that cutting frequency significantly altered the aboveground biomass production in
urban lawns. This can be explained by the fact that canopy $CO_2$ assimilation is a function of the
amount of assimilating plant matter (Wohlfahrt et al., 2008b). Wohlfahrt et al. (2008a) showed
that when the green area index (GAI) of an alpine grassland exceeded 4 m$^2$ m$^{-2}$, the gross
primary production (GPP) decreased due to shading, but also due to plant phenology. Directly
after cutting (three cuts per season), their grassland had a GAI of 0.5-2 m$^2$ m$^{-2}$, while directly
before cutting it had a GAI > 6 m$^2$ m$^{-2}$. The meadow-like lawns in our study were only cut once,
which indicates that the period in which the GAI of the canopy exceeded the optimum for $CO_2$
assimilation was very long. In contrast, the GAI of the utility lawn remained relatively close to
the optimum throughout the entire growing period. Furthermore, Klimeš and Klimešová (2002)
found that frequent mowing promoted the dominance of efficiently regrowing plant species,
which might provide an additional explanation for the higher NPP in our utility lawns. Our
results are also in agreement with Kaye et al. (2005), who found five-fold higher aboveground



NPP in an urban lawn than in a short-grass steppe. However, the urban lawn in that study was
fertilised and irrigated, while the urban lawns in our study were not. In a long-term field
experiment on cutting frequency effects on grass yield, Kramberger et al. (2015) found the
lowest yield in plots with the highest cutting frequency (2-week intervals) and the highest yield
in plots with moderate to low cutting frequency (8- to 12week intervals). This is in contrast to
our results from Uppsala and Malmö, but in line with the results from Gothenburg, where we
found higher aboveground biomass in the meadow-like lawns. However, we are unable to
explain the much higher NPP of the meadow-like lawns in Kyrkbyn.

### 4.2 Effect of cutting frequency on soil organic carbon in relation to similar management contrasts

The higher aboveground NPP in the utility lawns had a significant positive effect on soil carbon.
This was expected, since the clippings were not removed and were thus able to contribute
directly to soil organic matter formation. For this reason, the results of our study are not directly
applicable to mown grasslands or leys, which are usually harvested. The responses of SOC to
management intensity in those systems are not well studied, but studies performed to date show
mixed results ranging from no effect (Kramberger et al., 2015) to significantly positive effects of
high cutting frequency (Ammann et al., 2007). In the latter case, the difference in SOC stocks
between intensively and extensively managed grassland was attributed to differences in N
fertilisation, which caused N deficiency and thus N mining in the extensive grassland, leading to
stronger mineralisation of stable organic matter. The effects of grazing intensity on SOC are
much better studied than the effects of mowing intensity. Both positive (Reeder et al., 2004;
Smoliak et al., 1972) and negative (Abril and Bucher, 2001; Su et al., 2005) effects of low
compared with high grazing intensity on SOC have been reported. However, many of the studies
reporting negative effects of intensive grazing refer to overgrazing in semiarid areas, which is
associated with strongly reduced vegetation cover and soil erosion. The actual effects seem to be
context-specific, as found in a global meta-analysis conducted by McSherry and Ritchie (2013).
The found positive correlation of difference in SOC and average clay content across sites has to
be interpreted with caveats, since a clear causality is not given. It is realistic, that more of the C
input is stabilised in clay-rich soils (Poeplau et al., 2015b). However, this correlation did not
hold within the three sites at each city, which indicates that the found correlation of clay and



difference in SOC, as well as of clay and difference in NPP across all sites might as well
resemble a random city effect.
Overall, our findings and those of previous studies (Christopher and Lal, 2007; Poeplau et al.,
2015a) confirm that plant input driven by NPP is the major driver for SOC dynamics. Root
carbon input is recognised as being of major importance for building up soil organic matter,
since a higher fraction of root-derived carbon is stabilised in the soil than in aboveground plant
material (Kätterer et al., 2011). In temperate grasslands, up to 70% of the total NPP is allocated
to roots and their exudates (Bolinder et al., 2007). However, in the present study, management
intensity did not significantly influence root biomass, indicating that root production was
relatively favoured in the meadow-like lawns. A similar finding has been reported in a study
which found higher root biomass under diverse swards than under conventional, intensively
managed ryegrass-clover pastures (McNally et al., 2015). Therefore, altered root production
could not explain observed differences in SOC stocks in our study. However, the proportion of
aboveground plant material stabilised in the soil has been estimated to be 13% in a Swedish
long-term agricultural field experiment (Andrén and Kätterer, 1997). Similar values, i.e. around
10%, have been reported in other studies (Lehtinen et al., 2014; Poeplau et al., 2015b). It can be
assumed that lawn clippings undergo slightly lower stabilisation than straw in agricultural
systems, due to the lack of mixing of residues with stabilising mineral soil particles (Wiesmeier
et al., 2014). The mean annual difference in SOC sequestration between the two lawn types we
studied was 120 kg C ha$^{-1}$yr$^{-1}$. Assuming a constant stabilisation rate of 10% across all sites, the
calculated difference in SOC sequestration due only to different amounts of recycled clippings
would have been 69 kg C ha$^{-1}$yr$^{-1}$, which is only slightly more than half the observed difference.
Several studies report accelerated root turnover in more intensively managed grassland (Klumpp
et al., 2009; Leifeld et al., 2015). However, accelerated root turnover could result in either more
or less root-derived SOC, depending on the effect on total root growth and exudations
throughout the year, which is difficult to investigate (Johnen and Sauerbeck, 1977).
Interestingly, the soil C:N ratio was significantly lower in the utility lawns than in the meadow-
like lawns, although neither system was fertilised and both were equally exposed to N
deposition. Furthermore, the proportion of N-fixing leguminous plants was higher in the utility
lawns than in the meadow-like lawns only in Gothenburg. This might indicate that nitrogen





cycling was more closed in the utility lawns. Potentially, more nitrogen is lost via leaching in the
meadow-like lawns, because N mineralisation and plant N demand occur asynchronously
(Dahlin et al., 2005). The peak in N mineralisation usually occurs around midsummer (Paz-
Ferreiro et al., 2012), which might be too late for plant uptake when the grass is not mown and
would lead to N losses from the system. Another pathway of N loss is ammonia ($NH_3$)
volatilisation, which increases in later development stages of the plant due to ontogenetic
changes in plant N metabolism (Morgan and Parton, 1989). Whitehead and Lockyer (1989)
showed 10% N losses from decomposing grass herbage by $NH_3$ volatilisation. The consequences
of N deficiency for SOC dynamics are twofold: i) decreased NPP and thus decreased carbon
input (Christopher and Lal, 2007) and ii) increased heterotrophic respiration due to N mining in
more recalcitrant organic matter (Ammann et al., 2007). In an incubation experiment, Kirkby et
al. (2014) showed that more aboveground residues were stabilised in the soil when nitrogen was
added. Thus, negative effects of lawn management on soil N storage can feed back onto SOC,
which might also explain a certain proportion of the observed differences in SOC.
4.3 Implications for urban soil management
During the past decade, several studies have investigated biogeochemical cycles in urban soils,
since their relevance for the global carbon cycle and as a fundamental ecological asset in an
urbanising world is becoming increasingly evident (Lehmann and Stahr, 2007; Lorenz and Lal,
2009). Compared with data on agricultural land with similarly textured soils in the surroundings
of the study sites extracted from a national soil inventory database, we found on average 55%
(utility lawns) and 35% (meadow-like lawns) higher SOC stocks in the lawns we investigated.
Furthermore, it has been found in several studies that urban soils have higher carbon stocks than
native soils in adjacent rural areas, which can be attributed in particular to more optimised, but
also resource-consuming, management, including fertilisation and irrigation (Kaye et al., 2005;
Pouyat et al., 2009). However, in the present study we were able to show that SOC storage in
urban lawns can be increased at comparatively low cost under temperate climate conditions by
optimising NPP and leaving residues on the lawn. Losses of carbon and nutrients are thereby
minimised. Milesi et al. (2005) used the BIOME-BGC model to compare different lawn
management scenarios and found that applying 73 kg N and recycling the clippings was more
efficient for SOC sequestration (+40%) than applying 146 kg N and removing the clippings. For



the sites in Uppsala, Wesström (2015) calculated that the management of utility lawns creates 54
kg ha$^{-1}$ yr$^{-1}$ more C emissions than the management of meadow-like lawns. Subtracting this value
from the annual difference in SOC sequestration that we found (120 kg C ha$^{-1}$ yr$^{-1}$), the utility
lawns in our study sequester a non-significant amount of 66 kg ha$^{-1}$ yr$^{-1}$ more carbon than the
meadow-like lawns. However, for a full greenhouse gas budget, the effects of lawn management
on other trace gases, primarily nitrous oxide ($N_2O$), have to be considered (Townsend-Small and
Czimczik, 2010). In that case management of the clippings will most likely play a key role, since
coverage of the soil with organic material increases soil moisture and the availability of labile
carbon but decreases soil oxygen, all of which favour $N_2O$ formation (Larsson et al., 1998;
Petersen et al., 2011).

**5 Conclusions**

This investigation of urban lawns in three Swedish cities showed that cutting frequency alone
can exert a significant influence on soil carbon, mainly by increasing net primary production and
thus carbon inputs. However, this is most likely only true when cuttings are left on the lawn,
since belowground production did not show any differential response to cutting frequency.
Moreover, the observed difference in soil carbon could not be fully explained by the expected
stabilisation of aboveground-derived carbon input differences, which might denote that either
root-derived carbon dynamics or nitrogen mining also play an important role. If clippings are left
on the lawn, nitrous oxide emissions might comprise a significant fraction of the greenhouse gas
budget of lawns and have to be accounted for to judge the climate mitigation potential of
contrasting lawn or grassland management strategies.

**Acknowledgements**

This study was funded by Formas, the Swedish Research Council for Environment, Agricultural
Sciences and Spatial Planning (contract 225-2012-1369).





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





Table 1: Site characterisation with year of establishment, clay, silt and sand content [%], soil pH for utility lawns (U) and meadow-like lawns (M) and mean annual temperature [MAT, °C] and mean annual precipitation [MAP, mm] (1961-1990) for all three Swedish cities studied

| City | Site | Age | MAT | MAP | Clay U | Clay M | Silt U | Silt M | Sand U | Sand M | pH U | pH M |
|------|------|-----|-----|-----|----|----|----|----|----|----|----|----|
| Uppsala | Eriksberg | 1949 | 5.5 | 527 | 36 | 46 | 43 | 44 | 21 | 10 | ~6 | ~6 |
| | Sala backe | 1950 | | | 45 | 45 | 47 | 51 | 8 | 4 | ~6 | ~6 |
| | Tuna backar | 1951 | | | 33 | 23 | 47 | 45 | 20 | 32 | ~6 | ~6 |
| Malmö | Kirseberg | 1950 | 8.4 | 540 | 12 | 10 | 49 | 46 | 39 | 45 | 7.2 | 7.2 |
| | Sibbarp | 1953 | | | 15 | 15 | 48 | 47 | 38 | 38 | 7.4 | 7.8 |
| | Augustenborg | 1952 | | | 13 | 10 | 49 | 45 | 38 | 45 | 7.4 | 7.7 |
| Gothenburg | Guldheden | 1950 | 7.4 | 714 | 16 | 14 | 45 | 44 | 39 | 42 | 5.5 | 5.4 |
| | Kyrkbyn | 1955 | | | 16 | 22 | 62 | 55 | 21 | 23 | 5.8 | 5.7 |
| | Björkekärr | 1950 | | | 14 | 16 | 49 | 58 | 37 | 27 | 5.5 | 5.7 |

[x]year only approximate. *pH values for the Uppsala sites were not measured, and the values shown are estimates based on typical values for soils in Uppsala (e.g. Kätterer et al., 2011)

Table 2: Soil bulk density (BD) [g cm$^{-3}$] and SOC stocks [Mg ha$^{-1}$] according to equation 3. Standard deviation is given in brackets

| City | Site | Utility lawn BD | | Meadow-like lawn BD | | Utility lawn SOC stock | | Meadow-like lawn SOC stock | |
|------|------|------|------|------|------|------|------|------|------|
| Uppsala | Eriksberg | 1.13 | (0.04) | 1.13 | (0.16) | 74.8 | (11.4) | 63.1 | (8.7) |
| | Sala Backe | 1.14 | (0.03) | 1.1 | (0.07) | 96.2 | (9.3) | 69.8 | (24.2) |
| | Tuna Backar | 1.15 | (0.07) | 1.21 | (0.06) | 72.4 | (13.3) | 47.6 | (19.5) |
| Malmö | Kirseberg | 1.03 | (0.07) | 1.02 | (0.08) | 69.4 | (4.5) | 52.7 | (0.95) |
| | Sibbarp | 1.04 | (0.06) | 0.98 | (0.06) | 75 | (8.3) | 96.4 | (3.5) |
| | Augustenborg | 1.03 | (0.06) | 1.18 | (0.15) | 59.1 | (9.4) | 50.3 | (22.4) |
| Gothenburg | Guldhelden | 0.87 | (0.14) | 0.88 | (0.21) | 86.2 | (2.3) | 78.4 | (21.3) |
| | Kyrkbyn | 0.99 | (0.09) | 0.88 | (0.06) | 68.2 | (8.1) | 77.9 | (7.8) |
| | Björkekärr | 0.96 | (0.1) | 0.99 | (0.08) | 67 | (14.4) | 61.2 | (3.1) |




Figure captions:
Figure 1: Example of the vegetation model (equation 1) fit to a calculated cumulative growth
curve for a utility lawn in Björkekärr, Gothenburg.

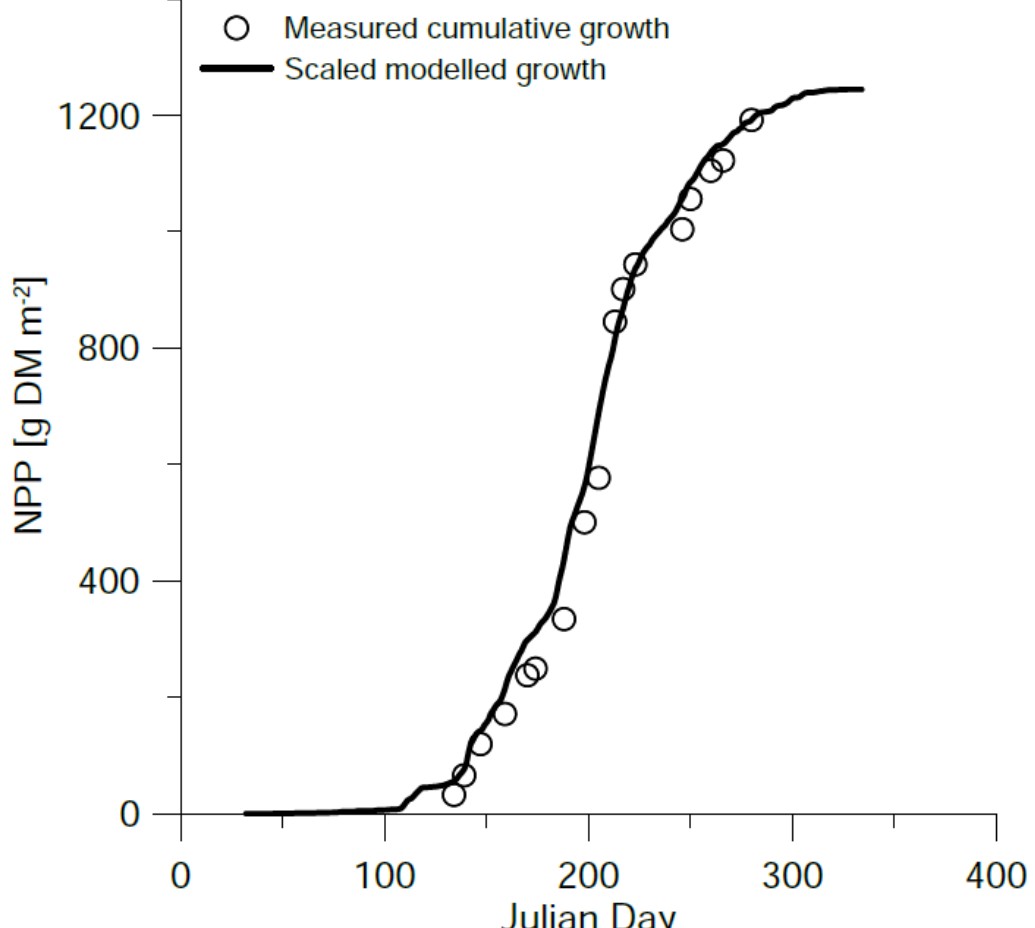








Figure 2: Bar plot showing estimated aboveground net primary production (NPP) of the two
different lawn types at each site. Errors bars indicate standard deviation and stars indicate
significant difference between treatments at the specific site (p<0.05).

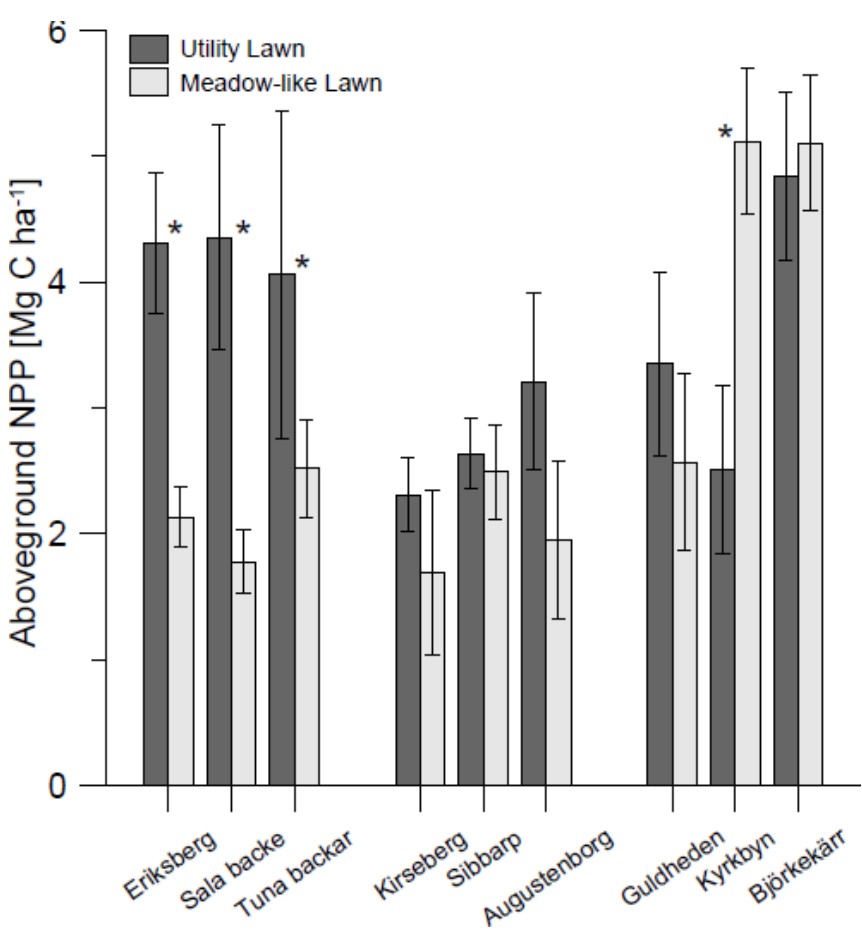










Figure 3: Bar plot showing total root biomass at 0-10 cm depth for the two different lawn types
at the sites in Uppsala. Errors bars indicate standard deviation and stars indicate significant
difference between treatments at the specific site ($p < 0.05$).

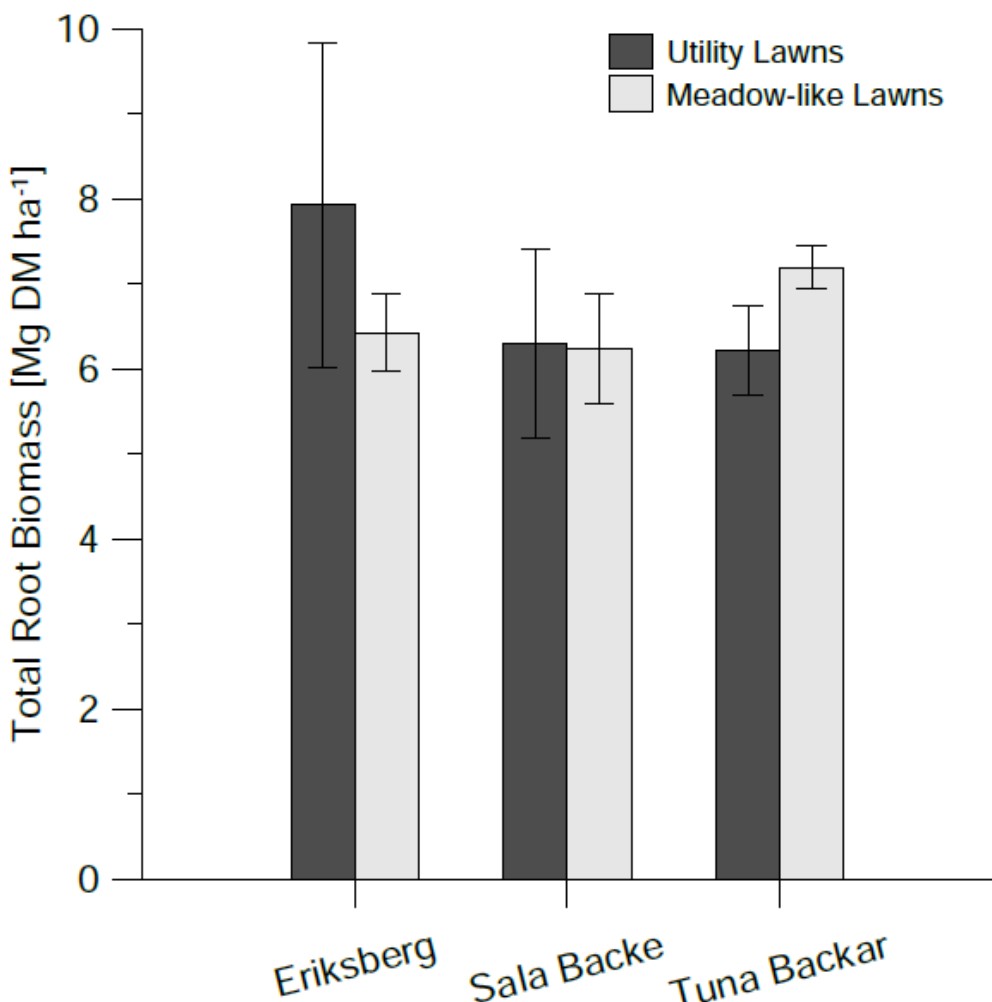







Figure 4: Bar plot showing measured soil organic carbon (SOC) concentration in the two
different lawn types at each site. Error bars indicate standard deviation and stars indicate
significant difference between treatments at the specific site ($p < 0.05$).

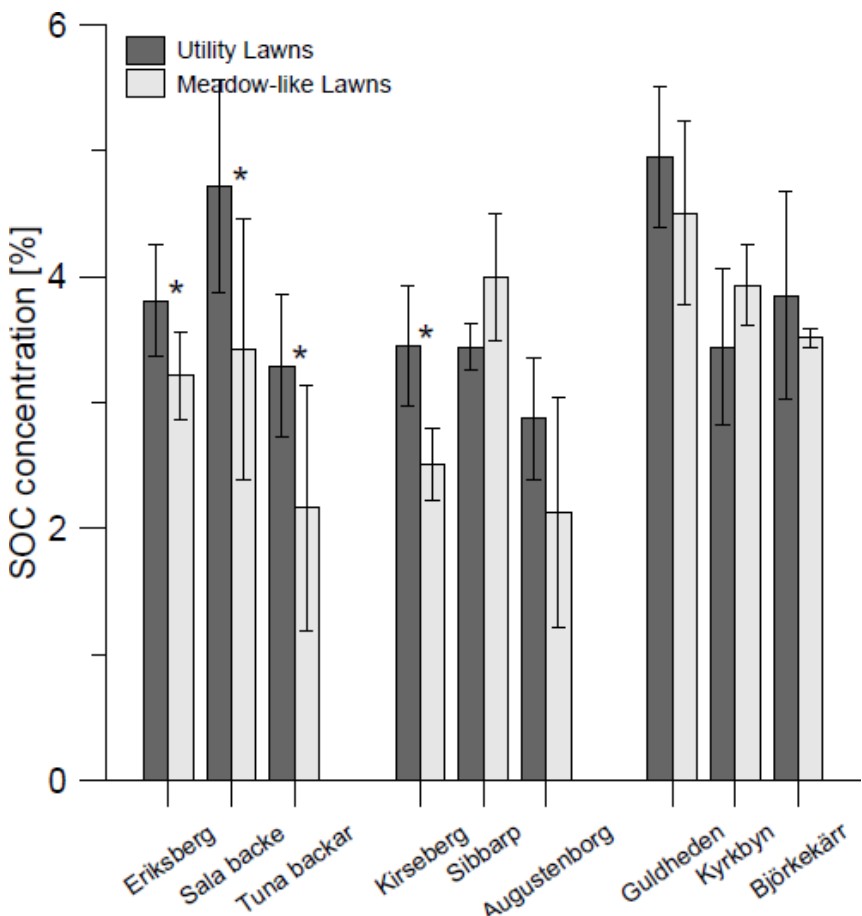









Figure 5: Difference in soil organic carbon (SOC) stock between utility and meadow-like lawns
as a function of difference in aboveground NPP for all sites.

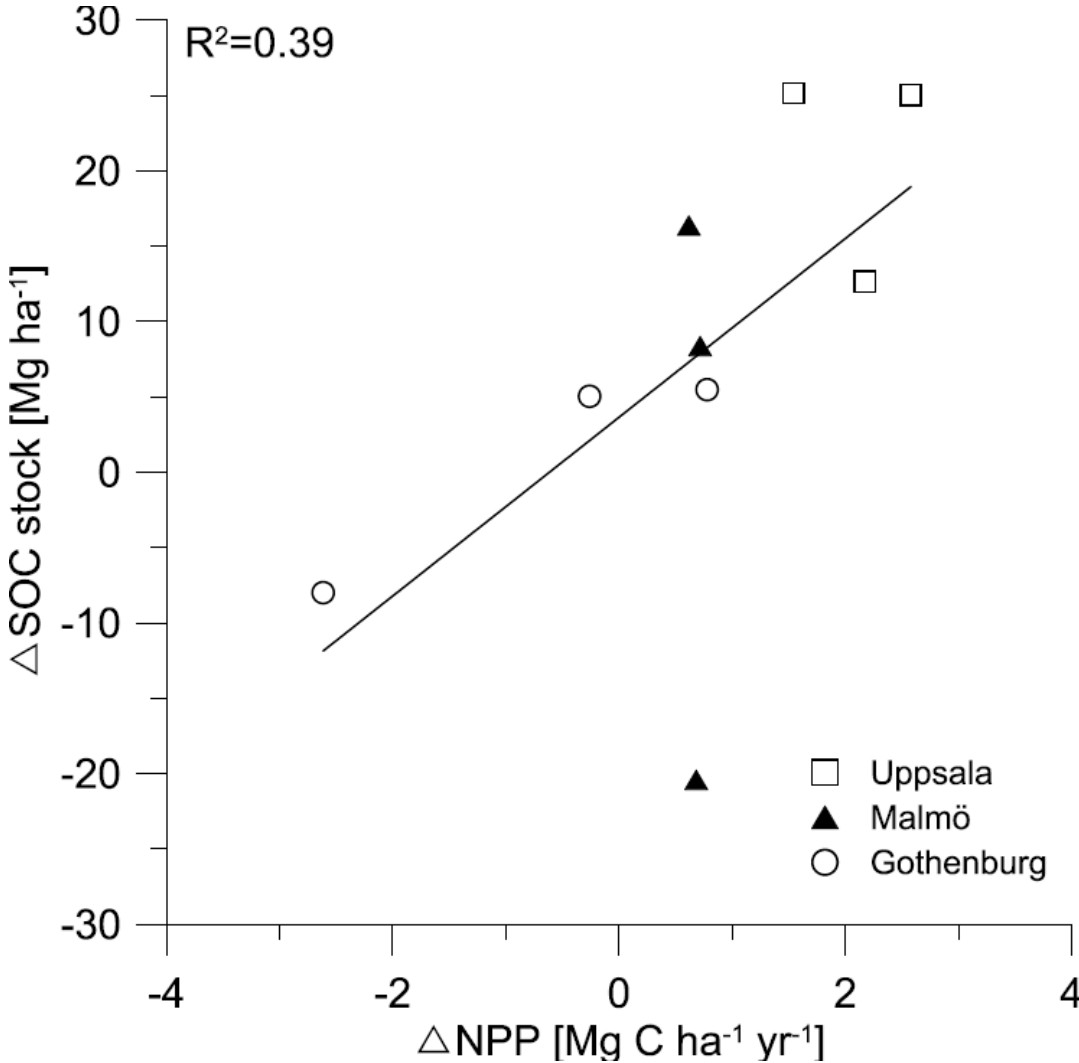










Figure 6: Bar plot showing measured C:N ratio of the two different lawn types at each site. Error
bars indicate standard deviation and stars indicate significant difference between treatments at
the specific site (p<0.05).

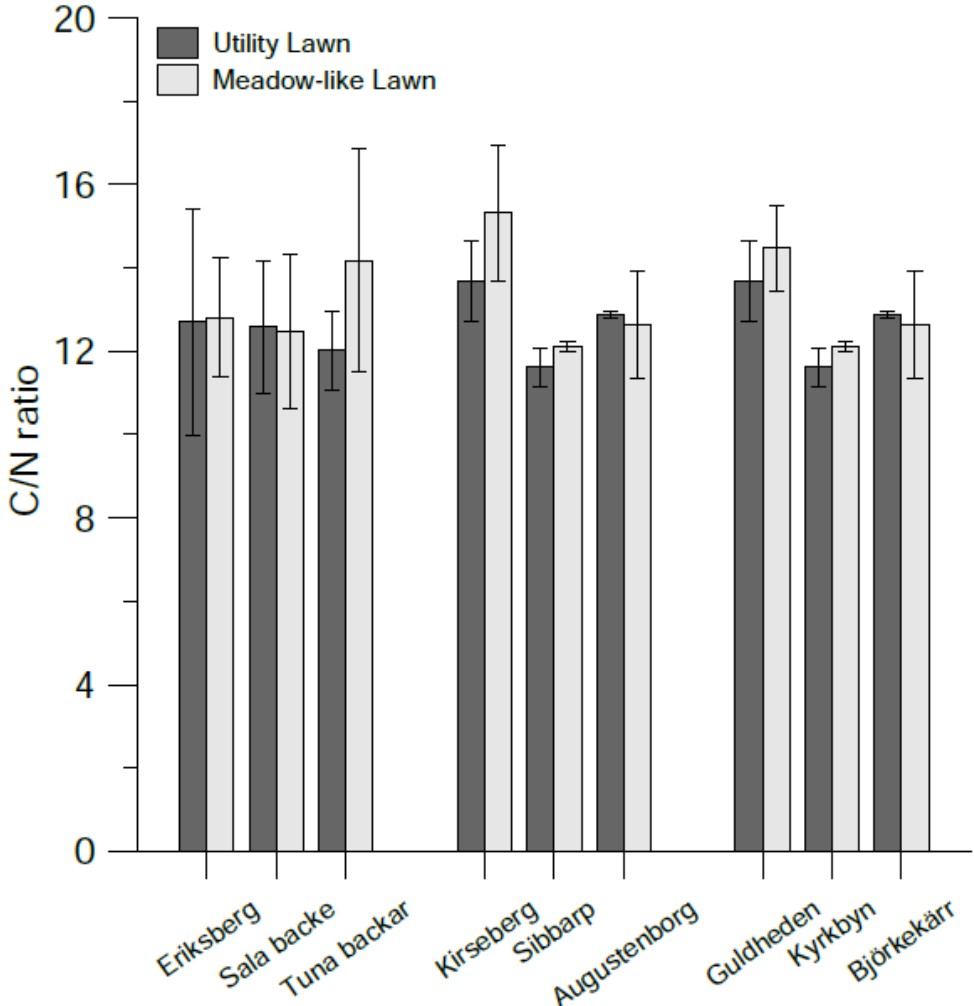
