# Peer review of "Effect of grassland cutting frequency on soil carbon storage A case study on public lawns in three Swedish cities"

_SOIL, 2015_

## Referee Comment (RC1) · Anonymous Referee #1 · 26 Feb 2016

The article titled: Effect of grassland cutting frequency on soil carbon storage- A case study on public lawns in three Swedish cites has essential prerequisite to be published in SOIL. In addition, this article displays the necessity to research the different sustainable management practices in grassland in relation with C and N stocks. Since, the most of grassland studies have been assessed in financial term but sometimes the functions of soil like C and N storage had been forgotten. In addition, Poeplau et al. include public areas like experimental sites demonstrating the originality and functionality of this study. Nowadays, is important consider too the urban public areas from the different countries because they have environmental properties which are contributing in the soil C and N storage. Although the part of material and methods, the authors

considered in the line 172: "...basic assumption was that the underlying pedology and initial soil carbon stocks were similar..." I think would be necessary at least do a mention about the soil types in every study site because is essential when you work with soil. On the other hand, I believe that in the line 210-211 the correlation between SOC stock and clay content is down. Is possible that the different studied site has different clay mineralogy?. In a recent study from Han et al., 2015 showed that in the most of cases the clay mineralogy is a better control factors in the content and stabilization SOC than clay content. Or maybe there are other soil related parameters? Is possible in the future the implementation of new grassland management practices like reduced tillage (1 time a year) together with a more cut times to incorporate part of the organic matter and so decrease the N2O formation?or other management practices?

Finally, I also think that the part about resutls is very short related to the disccusion part.

---

## Referee Comment (RC2) · Anonymous Referee #2 · 27 Feb 2016

Overall, this paper addresses an important topic (the effects of management on soil carbon storage in urban lawns). The authors make a good case for the use of urban lawns as a study system, and bring important attention to the importance of urban areas in biogeochemical cycles and sustainability. The authors find that frequent cutting can increase soil organic carbon stocks, which they attribute to increased NPP. These results will be of general interest to soil ecologists, as well as those interested in urban ecology and sustainability.

While the paper addresses an important topic, and the overarching story appears sound, there are several areas where revision would greatly improve this manuscript.

General comments:

The introduction to this paper would benefit from a thorough revision for structure and clarity. This includes breaking up the long initial paragraph, clearly defining important terms/concepts throughout, and ensuring statements are backed by adequate evidence (see specific comments). Additionally, while a significant focus of the introduction is focused on the importance of soil carbon and grassland carbon generally, and on agricultural systems, the discussion focuses largely on the importance of urban soils and their role in urban sustainability. More closely matching the tone and scope of the introduction to that of the discussion would greatly benefit the manuscript as a whole.

In the methods, ensure that the study design is clearly and adequately described, as well as the sampling regime. The authors mention "cities", "areas", "sites", "plots", etc. throughout and it can sometimes be unclear exactly what unit of analysis is being referred to. A figure showing the sampling design (including cities, sites, and plots) would be very beneficial for the reader, here. The frequency of sampling is also unclear, and it is important that this is clarified.

The "statistics" section is far too short, and insufficiently detailed to explain what was analyses were done. For example, in this case, I would expect the authors to report: what diagnostic tests were performed on the models, whether any variables were transformed for analysis, the error distribution used for the generalized linear models, dependent and independent variables used in multiple regression, the specific R packages used to conduct the analysis (e.g. lme4 vs. nlme, or other, for mixed effects models?), as well as how p-values were obtained for pairwise comparisons presented in the figures, etc. I advise that the authors review papers conducting similar analyses to determine the appropriate level of detail for this section and thoroughly revise it.

With regards to root biomass and the influence of roots on soil carbon, the authors make some fairly broad generalizations from a relatively minor sampling effort (only the top 10cm were sampled at a third of sites, and only at one time point). Root biomass sampled at one time may not be representative of increased root growth throughout the season (e.g. see Ziter and MacDougall 2013), and changes in root depth may alter

soil organic carbon. A more nuanced discussion of what can and cannot be inferred from this limited root data is warranted here.

Specific comments:

Abstract: Line 16-17: By "provide a high diversity" do you mean species diversity? Diversity of grassland types? Diversity in lawn management? This sentence is a bit awkward/unclear overall, and should be rephrased for clarity.

Introduction: Line 43: "demonstrated, that" remove comma. Line 44: use of "appropriation" is unclear here – do you mean all human use of NPP? Line 45/46: This sentence is quite vague, please clarify (the effects of different management types? Appropriation of NPP? Line 46: You move somewhat abruptly from general importance of soil C and grassland soil C, to the impact of cutting. Consider starting a new paragraph here? This would help break up your first paragraph, which is quite long. Line 56-60: This is a very long sentence with several ideas. Try and break this down into clearer, shorter sentences. Line 69: Depending on the previous land-use history, urban SOC stocks may continue to change over time – can you necessarily assume equilibrium in all cases? May want to consider whether this assumption holds across different geographical areas/types of cities? (e.g. see Raciti et al. 2011, Golubiewski 2006) Line 70: What do you mean by "functional types" of grasslands? Line 72: Define "lawn". Does your use of lawn include unmowed meadows/semi-natural grasslands in cities, or only managed/turfgrass areas? Line 80-81: Is this work part of the LAWN project? It is a bit unclear to me why the LAWN project is brought up in this context.

Methods: Line 93: Unclear what you mean by "in each area, triplicate plots of two different lawn types...". Does "area" refer to city here, or site? E.g. do you mean in each city, 3 sites were established, each site containing one pair of plots (one each of meadow/utility)? Or that within each site, three different plots of each lawn type were established (e.g. 6 total plots per site)? Make sure your study design is clearly described. Line 94: Give some indication (average, range, etc.) of how large the lawns
are in this study. Are plot sizes relatively equal? Highly unequal? If unequal, how did you account for this in your sampling? Line 106: How often was sampling repeated? Was sampling only done after the first mowing, or after each subsequent mowing (in utility lawns)? Line 111: Were sampling locations chosen randomly? Haphazardly? Methodically? Also, were sampling locations consistent with repeated sampling, or did they differ? Line 156: How were 10cm depth bulk density soils taken to an approximate depth of 5-15cm? It is unclear to me what the 5-15cm depth is referring to. Line 174: Was soil texture measured in this study specifically? It is unclear where the soil texture values in Table 1 are from.

Results: Line 210: This sentence is confusingly worded, please rephrase.

Discussion: Line 272-73: See above comments re: root production. Without temporal sampling, it is difficult to assess whether roots could have made a larger contribution that was not detected at a single time point. Line 278-304: ThisS entire paragraph focuses on the role of N in your system, however N is rarely mentioned in the introduction, nor is it included in the study focus at the end of the introduction. As N is an important part of the discussion, consider introducing this topic earlier in the paper. Section 4.3 (Implications for urban soil management): This is an important contribution of your paper. Additional work to consider here is be that of Jill Edmondson, who has published several interesting recent papers regarding urban soil carbon, that would be informative in the context of your work.

Figures and Tables: Please ensure that all axes are clearly labeled (e.g. "Study Site" is missing from the x axis in several figures). In the figure legends, the importance of "stars" is indicated in several figures for which there are no stars present. Ensure that the figure legend clearly describes the figure at hand, and is tailored to each individual figure.

---

## Author Comment (AC1) · 9 Mar 2016

The article titled: Effect of grassland cutting frequency on soil carbon storage- A case study on public lawns in three Swedish cites has essential prerequisite to be published in SOIL. In addition, this article displays the necessity to research the different sustainable management practices in grassland in relation with C and N stocks. Since the most of grassland studies have been assessed in financial term but sometimes the functions of soil like C and N storage had been forgotten. In addition, Poeplau et al. include public areas like experimental sites demonstrating the originality and functionality of this study. Nowadays, is important consider too the urban public areas from the different countries because they have environmental properties which are contributing

in the soil C and N storage. Although the part of material and methods, the authors considered in the line 172: "basic assumption was that the underlying pedology and initial soil carbon stocks were similar..." I think would be necessary at least do a mention about the soil types in every study site because is essential when you work withsoil.

We thank the reviewer for the very positive review. Unfortunatly, the soil type has not been determined in this study, since most of the field work was conducted by technicians and students. However, regarding soil carbon sequestration, the measured parameters such as soil texture and pH should have a more important influence than soil type.

On the other hand, I believe that in the line 210-211 the correlation between SOC stock and clay content is down. Is possible that the different studied site has different clay mineralogy?. In a recent study from Han et al., 2015 showed that in the most of cases the clay mineralogy is a better control factors in the content and stabilization SOC than clay content. Or maybe there are other soil related parameters?

Unfortunately, clay mineralogy is not either available. As mentioned above, we have used soil texture and pH, as well as climate variables as explanatory variables. No other parameters are available. Furthermore, we detected a mistake regarding the clay effect: In the first review of the manuscript, the editor commented that we should test wether the differences in clay content were driving the observed SOC stock differences, which we have done. During this analysis (in the previous version), we also found that average clay content (of the site) was correlated to SOC stock difference (R2=0.26) and included it in the manuscript. However, before it did show in the generalized linear model, which we however did not question at that stage. So, clay content (as explanatory variable in the model) was not significant. We have changed the sentence accordingly, which now reads: "Although clay content did not improve the model fit of the generalized linear model, difference in SOC stock did also increase with average clay content ($R^2$=0.26, ns)."

Is possible in the future the implementation of new grassland management practices like reduced tillage (1 time a year) together with a more cut times to incorporate part of the organic matter and so decrease the N2O formation?or other management practices?

In urban systems, grassland renovation tillage is obviously not a practical option, since this would destroy the lawn. Also, such renovation tillage has been found to increase N2O emissions (short-term) due to the rapid mineralization of organic matter and thus release of nitrogen and shortage of oxygen (e.g. Velthof 2009, Nutrient cycling in agroecosystems). But those studies are mostly not conducted in mulched systems, but in harvested systems. Therefore, the effect of tillage in such a mulched system as investigated here is highly uncertain, but might in fact be positive for SOC stocks (due to the higher stabilization of clippings. However, as mentioned in the next comment, the discussion is already quite long as compared to the results section, therefore we decided not to enlarge it with some speculations on how (in agricultural systems) the greenhouse gas budget of managed grasslands can be further improved.

Finally, I also think that the part about results is very short related to the discussion part.

We agree, but we do not see why this should be a problem here. We consider the result section to be complete.

---

## Author Comment (AC2) · 9 Mar 2016

Overall, this paper addresses an important topic (the effects of management on soil carbon storage in urban lawns). The authors make a good case for the use of urban lawns as a study system, and bring important attention to the importance of urban areas in biogeochemical cycles and sustainability. The authors find that frequent cutting can increase soil organic carbon stocks, which they attribute to increased NPP. These results will be of general interest to soil ecologists, as well as those interested in urban ecology and sustainability. While the paper addresses an important topic, and the overarching story appears sound, there are several areas where revision would greatly improve this manuscript.

General comments:

The introduction to this paper would benefit from a thorough revision for structure and clarity. This includes breaking up the long initial paragraph, clearly defining important terms/concepts throughout, and ensuring statements are backed by adequate evidence (see specific comments). Additionally, while a significant focus of the introduction is focused on the importance of soil carbon and grassland carbon generally, and on agricultural systems, the discussion focuses largely on the importance of urban soils and their role in urban sustainability. More closely matching the tone and scope of the introduction to that of the discussion would greatly benefit the manuscript as a whole.

We agree, that the introduction did benefit from restructuring that has been performed in the revised version. We now structured it into 5 separate paragraphs, which roughly contain the following: 1. Importance of soil C and grasslands 2. Grassland management and soil C (what do we know) 3. Lack of data due to lack of good study systems- urban lawns are good study systems! 4. Importance of urban lawns 5. Scope of the study- management in urban lawns

We see that the reviewer criticizes, that the introduction is quite general (including agricultural grasslands...), while the study has been conducted in a very special environment, on which not much is known. However, in contrast to many other studies on urban soils, the scope here is not a comparison to non-urban soils, but is a comparison of two management options. Those could potentially also appear in agriculture or other contexts, so the introduction is used to clarify that the set-up is ideal to answer our main question, which adds to our general knowledge of management effects (cutting frequency) on SOC storage. In our view, this is also the main topic of the discussion section, which ends in a last paragraph (4.3) that is more related to urban soils. Therefore we think that the tone and scope of those two sections do not differ too much. However, the reviewer has suggested a couple of nice studies in urban soils we were not aware of. We have rewritten and expanded the paragraph 4 of the introduction section and think that the scope of the paper now became somewhat more "urban".

In the methods, ensure that the study design is clearly and adequately described, as well as the sampling regime. The authors mention "cities", "areas", "sites", "plots", etc. throughout and it can sometimes be unclear exactly what unit of analysis is being referred to. A figure showing the sampling design (including cities, sites, and plots) would be very beneficial for the reader, here. The frequency of sampling is also unclear, and it is important that this is clarified.

After the revision (see also response to specific comments), we think that this point became quite clear (3 plots of each management type at each of 3 sites in each of 3 cities). Given the fact, that the paper has already 6 figures, we would prefer to leave this rather "trivial" graph and hope that the design is understandable without. Biomass sampling was conducted after each mowing event. We specified that in the revised version.

The "statistics" section is far too short, and insufficiently detailed to explain what was analyses were done. For example, in this case, I would expect the authors to report: what diagnostic tests were performed on the models, whether any variables were transformed for analysis, the error distribution used for the generalized linear models, dependent and independent variables used in multiple regression, the specific R packages used to conduct the analysis (e.g. lme4 vs. nlme, or other, for mixed effects models?), as well as how p-values were obtained for pairwise comparisons presented in the figures, etc. I advise that the authors review papers conducting similar analyses to determine the appropriate level of detail for this section and thoroughly revise it.

We revised the whole section and hope that the level of detail is now appropriate. The section reads as follows: "All statistical analyses were performed with the R software version 3.1.2. We used linear mixed effect models to analyse the effect of lawn management on aboveground NPP and SOC concentration and stocks using the R Package nlme (Pinheiro et al., 2009). Management (utility vs. meadow-like lawn) was used as the fixed effect, while city and site were used as nested random effects (site nested in city). We used Tukey-type multiple comparison Post-Hoc analysis (R Package multcomp) to test the management effect at each site for significance (p<0.05). Average differences in SOC stocks between the different lawn types at each site (dependant variable) were calculated and related to different explanatory variables (independent variables), such as average clay content, differences in clay content between lawn types (absolute and relative), soil pH, mean annual temperature (MAT), mean annual precipitation (MAP) and differences in aboveground NPP. Generalised linear models with Gaussian error distribution were used for multiple regression analysis. Model performance was evaluated using the Akaike Information Criterion (AIC). The variable "clay content" had to be transformed to approximate normal distribution. For both model approaches (mixed effect model and generalized linear model) we used residual plots to study whether i) the regression function was linear, ii) the error terms had constant variance, iii) the error terms were independent, iv) there were outliers or v) the error terms were normally distributed. All values in the text and diagrams represent mean±standard deviation."

With regards to root biomass and the influence of roots on soil carbon, the authors make some fairly broad generalizations from a relatively minor sampling effort (only the top 10cm were sampled at a third of sites, and only at one time point). Root biomass sampled at one time may not be representative of increased root growth throughout the season (e.g. see Ziter and MacDougall 2013), and changes in root depth may alter soil organic carbon. A more nuanced discussion of what can and cannot be inferred from this limited root data is warranted here.

We agree and added the following sentences: "However, the informative value of the obtained root data is certainly limited, since root biomass was only determined in one city, to a depth of 10 cm and at one point in time. It can thus not be assumed that the measured root biomass measured is representative for root growth throughout the season (Ziter and MacDougall, 2013). Furthermore, potential management effects on the depth distribution of belowground biomass cannot be inferred."

Specific comments: Abstract: Line 16-17: By "provide a high diversity" do you mean

species diversity? Diversity of grassland types? Diversity in lawn management? This sentence is a bit awkward/unclear overall, and should be rephrased for clarity.

We agree. We were actually referring to a high functional diversity and changed the sentence as follows: "City lawns constitute a unique study system in this context, since they provide a high functional diversity and thus a wide range of different management intensities per unit area.

Introduction: Line 43: "demonstrated, that" remove comma. Line 44: use of "appropriation" is unclear here – do you mean all human use of NPP?

We removed the commas and added (harvest) to clarify.

Line 45/46: This sentenceis quite vague, please clarify (the effects of different management types? Appropria-tion of NPP?

We actually decided to delete this sentence, since it was meant as some kind of bridge between the last and the next sentence, but now we started a new paragraph instead.

Line 46: You move somewhat abruptly from general importance of soilC and grassland soil C, to the impact of cutting. Consider starting a new paragraph here? This would help break up your first paragraph, which is quite long.

We agree and changed that accordingly.

Line 56-60: This is a very long sentence with several ideas. Try and break this down into clearer, shorter sentences.

We have changed the sentence as follows: "Management effect on SOC are presumably smaller than land use change effects such as conversion from permanent pasture to arable land (Poeplau and Don, 2013) and might not be visible in the short term. To assess those changes, it is therefore important to find suitable study systems with long-lasting strong contrasts in management intensity over a limited spatial scale and with limited soil variability."

Line 69: Depending on the previous land-use history, urban SOC stocks may continue to change over time – can you necessarily assume equilibrium in all cases? May want to consider whether this assumption holds across different geographical areas/types of cities? (e.g. see Raciti et al. 2011, Golubiewski 2006).

Thank you for mentioning two interesting studies, that we did not consider before. We agree and changed the sentences as follows: "Lawns, public green areas and parks are omnipresent in urban areas and are usually managed in a similar way for a long time, so that depending on the prior land-use type equilibrium SOC stocks might be approximated (Raciti et al., 2011)."

Line 70: What do you mean by "functional types" of grasslands?

We changed the sentence, which now reads as follows: "Over a comparatively small spatial scale, a wide range of different management intensities can be present."

Line 72: Define "lawn". Does your use of lawn include unmowed meadows/semi-natural grasslands in cities, or only managed/turfgrass areas?

Here, we were indeed only including turfgrass lawns and therefore added the word turfgrass (as in the following sentences).

Line 80-81: Is this work part of the LAWN project? It is a bit unclear to me why the LAWN project is brought up in this context.

We agree that this was not clear. We added in the next sentence: "..., as part of the lawn project,..." and think that it might be of interest for the reader i) in which context this work has been performed and ii) what other work has been done in this project.

Methods: Line 93: Unclear what you mean by "in each area, triplicate plots of twod-ifferent lawn types". Does "area" refer to city here, or site? E.g. do you mean in each city, 3 sites were established, each site containing one pair of plots (one each of meadow/utility)? Or that within each site, three different plots of each lawn type were established (e.g. 6 total plots per site)? Make sure your study design is clearly

described.

We agree that this was confusing, but the latter was true. In each city (n=3), we established 3 sites and at each site three different plots of each lawn type were established. We have now replaced "location" and "area" with "site", as it is also called in the tables.

Line 94: Give some indication (average, range, etc.) of how large the lawns are in this study. Are plot sizes relatively equal? Highly unequal? If unequal, how did you account for this in your sampling?

We added the following sentences: "The size of the individual lawns was highly unequal with a range of 0.05-2.5 ha due to the heterogeneity of urban landscapes. To obtain representative average values for the whole individual lawn, we conducted all samplings described below adjusted to the size of the lawn, instead of using a "fixed grid"."

Line 106: How often was sampling repeated? Was sampling only done after the first mowing, or after each subsequent mowing (in utility lawns)?

We changed the sentence: ". . .after each mowing event."

Line 111: Were sampling locations chosen randomly? Haphazardly? Methodically? Also, were sampling locations consistent with repeated sampling, or did they differ?

It was done methodically by selecting "representative locations", or excluding any special situations (under trees, in sinks. . .), which is described in the manuscript (l. 112-113). Sampling locations differed. We added the following sentence: "Repeated sampling was not conducted on the identical sampling locations."

Line 156: How were 10cm depth bulk density soils taken to an approximate depth of 5-15cm? It is unclear to me what the 5-15cm depth is referring to.

We rephrased the sentence, which should yield clarity: "Soil bulk density [g cm-3] was determined by taking undisturbed cylindrical soil cores of 7 cm diameter and 10 cm

height in an approximate soil depth of 5-15 cm, drying them at 105°C and weighing them."

Line 174: Was soil texture measured in this study specifically? It is unclear where the soil texture values in Table 1 are from.

Yes, soil texture was measured in this study. We have now added a two sentences to the M&M section: "Soil texture was determined with the pipette method according to ISO 11277. As a slight modification, wet sieving prior to sedimentation was done to 0.2 mm compared to 0.063 mm prescribed in the ISO method."

Results: Line 210: This sentence is confusingly worded, please rephrase.

We changed the sentence, which now reads as follows: "Although clay content did not improve the model fit of the generalized linear model, difference in SOC stock did also increase with average clay content (R2=0.26, ns)."

Discussion: Line 272-73: See above comments re: root production. Without temporal sampling, it is difficult to assess whether roots could have made a larger contribution that was not detected at a single time point.

See response above.

Line 278-304: This entire paragraph focuses on the role of N in your system, however N is rarely mentioned in the introduction, nor is it included in the study focus at the end of the introduction. As N is an important part of the discussion, consider introducing this topic earlier in the paper.

Without becoming to detailed on that, we mentioned before in the introduction, that nutrient status can interact with the C cycle in soils and cite some related studies (l.53-54). We therefore do not agree that it was not included at all. However, we agree that it is worth mentioning N in the study focus (at least the C:N ratio) which we did as follows: "We examined how cutting frequency affected: i) NPP, SOC and soil carbon to nitrogen ratio (C:N), and ii) the mechanisms involved for potential differences in SOC storage."

Section 4.3 (Implications for urban soil management): This is an important contribution of your paper. Additional work to consider here is be that of Jill Edmondson, who has published several interesting recent papers regarding urban soil carbon, that would be informative in the context of your work.

We agree and added one reference (Edmondson et al. 2012).

Figures and Tables: Please ensure that all axes are clearly labeled (e.g. "Study Site" is missing from the x axis in several figures).

We don't think that we have to add "study site", when the name of each study site is given in the axis and the legend describes that the different study sites are displayed.

In the figure legends, the importance of "stars" is indicated in several figures for which there are no stars present. Ensure that the figure legend clearly describes the figure at hand, and is tailored to each individual figure.

We have changed that accordingly by deleting the part of the sentence mentioning the stars.

---

## Author Response (AR1)

Dear Mrs. Boix-Fayos,

thank you very much for the overall positive evaluation of our manuscript. We have tried to solve the remaining two issues in the below described way and hope that the revised manuscript has now sufficient quality to be published.

1. We have now included the average CN ratio of each site in Table 1, as an indicator for organic matter quality and potential N supply. Unfortunately, no other soil parameter has been measured.

2. We added some sentences on Average SOC and NPP in each city as an inter-city comparison. We found SOC and NPP to be highly correlated in that comparison, added a new figure on that and concluded that SOC is also determining NPP to some extent. The folliwng sentences were added to the results: "The close link between NPP and SOC was also found when comparing city-average values of the two variables (Figure 6). The highest SOC concentrations and NPP values were found in Gothenburg. Interestingly, Gothenburg was also the city in which the difference between treatments was smallest for both parameters. This might reveal, that not only higher NPP leads to higher SOC, but that a presumably higher baseline SOC content can provide higher soil fertility and thus productivity even on extensively managed lawns."